# MEAL: A Benchmark for Continual Multi-Agent Reinforcement Learning

## Abstract

Benchmarks play a crucial role in the development and analysis of reinforcement learning (RL) algorithms, with environment availability strongly impacting research. One particularly underexplored intersection is continual learning (CL) in cooperative multi-agent settings. To remedy this, we introduce **MEAL** (**M**ulti-agent **E**nvironments for **A**daptive **L**earning), the first benchmark tailored for continual multi-agent reinforcement learning (CMARL). Existing CL benchmarks run environments on the CPU, leading to computational bottlenecks and limiting the length of task sequences. MEAL leverages JAX for GPU acceleration, enabling continual learning across sequences of up to 100 tasks on a standard desktop PC within a few hours. Evaluating popular CL and MARL methods reveals that naïvely combining them fails to preserve network plasticity or prevent catastrophic forgetting of cooperative behaviors.

## 1 Introduction

Continual RL has recently attracted growing interest [12, 6, 7, 10], but remains largely unexplored in multi-agent settings [31, 32]. Combining the two introduces unique challenges. In cooperative environments, agents must establish implicit conventions or roles for effective coordination [26]. As tasks or dynamics shift, these conventions can break down, making continual MARL significantly harder than its single-agent counterpart. Forgetting past partners or roles can cause the entire team to fail, amplifying the impact of catastrophic forgetting through inter-agent dependencies. Unlike traditional MARL, CMARL involves non-stationarity not only due to the presence of other learning agents, but also from a shifting task distribution [32]. This dual pressure demands agents that can generalize, adapt, and transfer knowledge more robustly than in standard single-agent continual or static multi-agent settings. This setting is relevant for applications where agents must adapt to evolving environments without forgetting prior coordination strategies. For instance, autonomous vehicles must navigate unseen roads, adapt to new traffic regulations, and interact with unfamiliar human drivers, while occasionally coordinating with other AVs. Similarly, warehouse robots deployed in a new facility must quickly adapt to different layouts and workflows, while preserving established collaborative behaviors.

To analyze how current methods handle the interplay between continual learning and multi-agent coordination, and to drive progress in this domain, we introduce **MEAL**, the first benchmark for CMARL. To the best of our knowledge, MEAL[1] is also the first continual RL library to leverage JAX for end-to-end GPU acceleration. Traditional CPU-based benchmarks are limited to short sequences (5–15 tasks) due to low environment throughput and task diversity [25, 21, 28], making them ill-suited for the computational demands of cooperative continual learning. MEAL's end-to-end

---

[1]The code and environments are accessible on Anonymous GitHub.

JAX pipeline removes this barrier, enabling training on up to 100 tasks within a few hours on a single desktop GPU. This unlocks new research directions for scalable, cooperative continual learning in resource-constrained settings.

MEAL is built on Overcooked [5], where agents are known to exploit spurious correlations in fixed layouts, leading to poor generalization even under minor changes [15]. As a result, a task sequence with small layout variations is sufficient to pose a challenging continual learning problem. Successfully learning across such a sequence requires agents to avoid layout-specific overfitting and instead develop robust, transferable coordination strategies. Sequential Overcooked layouts thus offer a controlled and reproducible way to introduce meaningful task diversity.

The **contributions** of our work are three-fold. (1) We introduce MEAL, the first CMARL benchmark, consisting of handcrafted and procedurally generated Overcooked environments spanning three difficulty levels. (2) We leverage JAX to build the first end-to-end GPU-accelerated task sequences for continual RL, enabling efficient training on low-budget setups. (3) We implement six popular CL methods in JAX and evaluate them in various MEALs, revealing key shortcomings in retaining cooperative behaviors and adapting to shifting roles across tasks.

## 2  Related Work

**Continual Reinforcement Learning (CRL)**    Continual reinforcement learning studies how agents can learn sequentially from a stream of tasks without forgetting previous knowledge. A wide range of methods have been proposed, including regularization-based approaches such as EWC [14], SI [33], and MAS [2]; architectural strategies such as PackNet [18]; and replay-based methods like RePR [3]. More recent works focus on scalability [12], memory efficiency [7], and stability during training [6]. However, these methods are almost exclusively developed for single-agent settings, and their behavior under multi-agent coordination remains largely unexplored.

**Multi-Agent Reinforcement Learning (MARL)**    In MARL, multiple agents learn to act in a shared environment, often with partial observability and either cooperative or competitive goals [13, 20]. A major focus has been on cooperative settings, where agents share a reward function and must learn to coordinate [17, 11]. Popular algorithms include IPPO [8], VDN [27], QMIX [22], and MAPPO [30]. Many benchmarks assume a static environment and fixed task, making them unsuitable for studying continual learning or transfer across environments.

**Benchmarks**    Standard CRL benchmarks include Continual World [29], COOM [28], and CORA [21]. While effective in single-agent settings, they either lack multi-agent capabilities or suffer from slow CPU-bound environments. For MARL, environments like SMAC [24], MPE [19], and Melting Pot [1] are widely used, but are not designed for continual evaluation. Overcooked [5] has emerged as a useful domain for studying coordination, with recent implementations in JAX [23]. Our benchmark builds on Overcooked and introduces procedural variation to create long task sequences for continual MARL.

**Overcooked**    The Overcooked environment [5] is a cooperative multi-agent benchmark inspired by the popular video game of the same name. Agents control chefs in a grid-based kitchen, coordinating to prepare and deliver dishes through sequences of interactions with environment objects such as pots, ingredient dispensers, plate stations, and delivery counters. The environment is designed to require both motion and strategy coordination, making it a standard testbed for evaluating collaborative behaviors.

Compared to the large state spaces and high agent counts in benchmarks like Melting Pot [1] and SMAC [24], Overcooked operates on small grid-based environments with only two agents. However, its complexity arises not from scale but from credit assignment challenges due to shared rewards, and the need for precise coordination, as agents must execute tightly coupled action sequences to complete tasks successfully [13]. However, its fully observable and symmetric setup reduces the need for explicit communication.

Table 1: Comparison of existing Reinforcement Learning benchmarks with MEAL

| Benchmark | No. Tasks | Difficulty Levels | GPU-accelerated | Action Space | Multi-Agent | Continual Learning |
|---|---|---|---|---|---|---|
| CORA [21] | 31 | ✗ | ✓ | Mixed | ✗ | ✓ |
| MPE [19] | 7 | ✗ | ✗ | Continuous | ✓ | ✗ |
| SMAC [24] | 14 | ✓ | ✗ | Discrete | ✓ | ✗ |
| Continual World [29] | 10 | ✗ | ✗ | Continuous | ✗ | ✓ |
| Melting Pot [1] | 49 | ✗ | ✗ | Discrete | ✓ | ✗ |
| Google Football [16] | 14 | ✓ | ✓ | Discrete | ✓ | ✗ |
| JaxMARL [23] | 33 | ✗ | ✓ | Mixed | ✓ | ✗ |
| COOM [28] | 8 | ✓ | ✗ | Discrete | ✗ | ✓ |
| **MEAL** | 25 | ✓ | ✓ | Discrete | ✓ | ✓ |

## 3  Preliminaries

**Cooperative Multi-Agent MDP**   We formulate the setting as a fully observable cooperative multi-agent task, modeled as a Markov game defined by the tuple $\langle N, S, A^i_{i \in N}, P, R, \gamma \rangle$, where $N$ is the number of agents, $S$ is the state space, $A^i$ is the action space of agent $i$ with joint action space $A = A^1 \times \cdots \times A^N$, $P : S \times A \times S \to [0,1]$ is the transition function, $R : S \times A \times S \to \mathbb{R}$ is a shared reward function, and $\gamma \in [0, 1)$ is the discount factor. In the fully observable setting, each agent receives the full state $s \in S$ at every time step.

**Continual MARL**   We consider a continual MARL setting in which a shared policy $\pi_\theta = \pi^i_{\theta\, i \in N}$ is learned over a sequence of tasks $\mathcal{T} = \mathcal{M}_1, \ldots, \mathcal{M}_T$, where each $\mathcal{M}_t = \langle N, S_t, A^i i \in N, P_t, R_t, \gamma \rangle$ is a fully observable cooperative Markov game with consistent action and observation spaces. At training phase $t$, agents interact exclusively with $\mathcal{M}_t$ for a fixed number of iterations $\Delta$, collecting trajectories $\tau_{t,1}, \ldots, \tau_{t,\Delta}$ to update their policy. Past tasks and data are inaccessible, and no joint training or replay is allowed. The focus of this work is on the task-incremental setting, where the task identity is known during training but hidden at evaluation. The objective is to maximize cumulative performance across all tasks and mitigate forgetting.

## 4  MEAL

We introduce MEAL, the first benchmark for CMARL, built on the JaxMARL [23] version of Overcooked. JAX [4] provides just-in-time compilation, automatic differentiation, and vectorization through XLA, enabling high-performance and accelerator-agnostic computation. We incorporate the original five layouts from Overcooked-AI [5] and design 20 additional handcrafted environments.

### 4.1  Environment Dynamics

**Observations**   Each agent receives a fully observable grid-based observation of shape $(H, W, 26)$, where $H$ and $W$ are the height and width of the environment, and the 26 channels encode entity types (e.g., walls, agents, onions, plates, pots, delivery stations) and object states (e.g., cooking progress, held item). To ensure compatibility across environments in a continual learning setting, we fix $H_{\max}$ and $W_{\max}$ to the largest layout size and pad smaller layouts with walls. Observations are then standardized to shape $(H_{\max}, W_{\max}, 26)$.

**Action Space**   At each timestep, both agents select one of six discrete actions from a shared action space $\mathcal{A} = \{\texttt{up}, \texttt{down}, \texttt{left}, \texttt{right}, \texttt{stay}, \texttt{interact}\}$. Movement actions translate the agent forward if the target tile is free (i.e., not a wall or occupied), while $\texttt{stay}$ maintains the current position. The $\texttt{interact}$ action is context-dependent and allows agents to pick up or place items, add ingredients to pots, serve completed dishes, or deliver them at the goal location. Importantly, there is no built-in communication action; all coordination emerges from environment interactions.

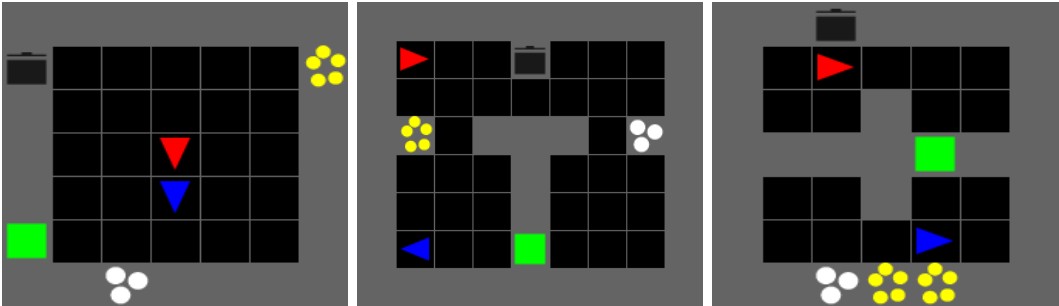

(a) Easy layouts are solvable by a single agent as long as key tiles remain accessible. Minimal coordination is required, and navigation is straightforward.

(b) Medium layouts contain bottlenecks that restrict movement and increase the likelihood of deadlocks. Agents must coordinate to avoid obstructing each other.

(c) Hard layouts split the map into disjoint regions, forcing agents to specialize. Solving the task requires deliberate cooperation and division of labor.

Figure 1: MEAL environments are grouped by layout difficulty: **easy** (minimal coordination), **medium** (bottlenecks and deadlocks), and **hard** (specialized cooperation due to partitioned access).

**Rewards**    Agents receive a shared team reward. The primary sparse reward is $+20$ for successfully delivering a completed soup. Optional shaped rewards can be added for partial task completion:

$$r_t = r_{\text{deliver}} + \alpha_1 \cdot \mathbb{1}_{\text{onion\_in\_pot}} + \alpha_2 \cdot \mathbb{1}_{\text{plate\_pickup}} + \alpha_3 \cdot \mathbb{1}_{\text{soup\_pickup}}, \qquad (1)$$

where $\alpha_1, \alpha_2, \alpha_3$ are reward shaping coefficients. All rewards are shared, encouraging cooperative behavior.

**Score Function**    Because MEAL environment layouts vary in size, the maximum achievable return differs per task. To ensure consistent comparison across sequences, we normalize returns using a reference performance: the average return of a converged IPPO agent trained from scratch on each environment. This normalizes the baseline score to 1. Note that scores can exceed 1 if a method generalizes or transfers better than the isolated baseline. We refer to this metric as the *IPPO-Normalized Score (INS)*.

## 4.2 Layout Difficulty

We categorize the handcrafted MEAL layouts into three levels of difficulty to better interpret agent behavior and learning dynamics. Appendix A depicts all the available MEAL layouts in difficulty groups. This grouping disentangles which coordination skills agents can acquire under varying structural constraints. Figure 1 illustrates representative layouts for each tier. In **easy** layouts, a single agent can often complete the task independently as long as key tiles remain unobstructed. **Medium** layouts introduce structural bottlenecks and tighter spatial constraints. Agents must coordinate their movement, such as implicit turn-taking in narrow passages, to sustain task throughput. **Hard** layouts partition the map into disjoint regions, often limiting each agent's access to only a subset of utilities (e.g., one agent sees only plates and onions). This forces agents to specialize and rely on their partner to complete the part of the recipe pipeline. Continual learning becomes especially challenging: agents must infer their new role based on the layout, and adapt strategies without forgetting past roles.

## 4.3 Task Sequences

Rather than a continuous domain shift, MEAL sequences involves discrete task boundaries, where agents transition between clearly distinct environments. This setup aligns with the task-incremental learning paradigm. We include three task sequence strategies.

**Ordered**    Tasks follow a fixed sequence. This setting enables controlled comparisons and rapid iteration during development, as the order remains constant across runs. Since the fixed task order reduces variance, fewer seeds are needed to draw reliable conclusions.

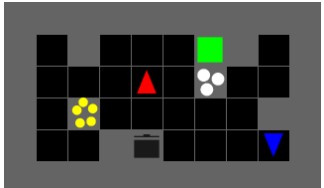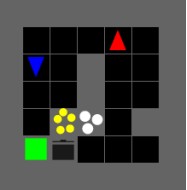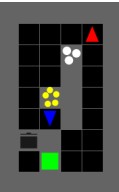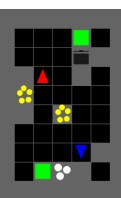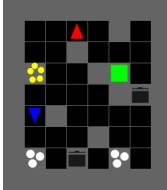

Figure 2: Five randomly generated Overcooked layouts. Each kitchen is guaranteed to be solvable.

**Random** To evaluate robustness, we sample task sequences randomly without replacement from the available pool. Since tasks differ substantially, the structure of the sequence has a strong impact on learning dynamics and knowledge transfer. Random ordering highlights method sensitivity to transferability between task pairs, and reflects the findings of Tomilin et al. [28] that performance often hinges on the characteristics of the first task and its downstream transfer potential.

**Generated** To support long sequences and continuous benchmarking, we procedurally generate new Overcooked layouts on the fly. Each layout is guaranteed to be solvable and varies in size, structure, and item placement. Figure 2 shows examples of generated environments. This setting offers a virtually infinite supply of tasks and evaluates true lifelong learning under continual exposure to unseen configurations.

## 4.4 Evaluation Metrics

We evaluate methods on three core metrics: **Average Performance**, **Forgetting**, and **Plasticity**. Let $s_i(j)$ denote the normalized score (see Section 4.1) on task $j$ after training on task $i$, and let the task sequence consist of $N$ tasks.

**Average Performance** We define average performance as the mean normalized score across all tasks at the end of training. This metric captures the balance between forward transfer and retention:

$$\mathcal{A} = \frac{1}{N} \sum_{i=1}^{N} s_N(i) \tag{2}$$

**Forgetting** Forgetting quantifies the degradation in performance on past tasks due to interference from training on later ones. For each task $i < N$, we compute the difference between the performance immediately after training and at the end of the sequence:

$$\mathcal{F} = \frac{1}{N-1} \sum_{i=1}^{N-1} \left( s_i(i) - s_N(i) \right) \tag{3}$$

**Plasticity** To evaluate continual training capacity over long task sequences, we measure the model's ability to fit new tasks under continual learning constraints. We compute the average training score (i.e., final performance on the current task right after training) across the entire sequence:

$$\mathcal{P} = \frac{1}{N} \sum_{i=1}^{N} s_i(i) \tag{4}$$

This isolates how quickly and effectively the method learns a new task under capacity constraints, independently of retention. Unlike Average Performance, Plasticity does not require evaluation on previously seen tasks.

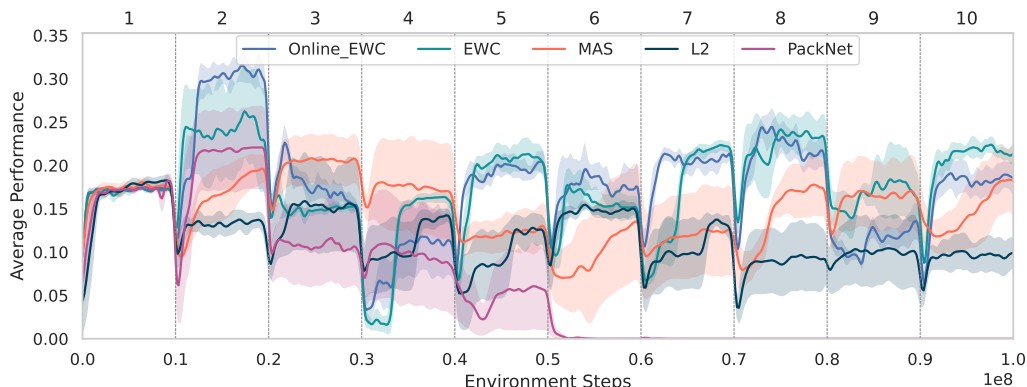

Figure 3: **Average performance** over the course of training on a 10-task sequence using the **Random** sequence strategy. Shaded regions indicate 95% confidence intervals across 5 seeds. Performance is measured as average normalized return across all tasks in the sequence.

## 5 Experiments

### 5.1 Setup

The agent is trained on each task $\mathcal{T}_i$ for $\Delta = 10^7$ environment steps on-policy. During training, we evaluate the policy after every 100 updates by running 10 evaluation episodes on all previously seen tasks. To ensure comparability across tasks with different layouts and reward scales, we normalize raw returns using a per-task transformation $f_i(\cdot)$, defined such that $f_i(score) = 0$ corresponds to a random agent and $f_i(score) = 1$ corresponds to a reference policy trained directly on task $\mathcal{T}_i$ until convergence (we use IPPO as the reference). This yields a unified measure of success across tasks.

We run each environment for 10 million environment steps using the random task selection strategy, repeated over five seeds. We leverage JAX to reduce the wall-clock time for training on a single environment to around 5 minutes. All experiments are conducted on a dedicated compute node with a 72-core 3.2 GHz AMD EPYC 7F72 CPU and a single NVIDIA A100 GPU. We adopt many of JAXMarl's default settings for our network configuration, IPPO setup, and training processes. For exact hyperparameters please refer to Appendix B.

### 5.2 Baselines

We evaluate several continual learning methods. Fine-tune (**FT**) is a naive baseline where the policy is trained sequentially across tasks without any mechanism to prevent forgetting. **L2-Regularization** [14] adds a penalty on parameter changes to encourage stability. **EWC** [14] is a regularization method that penalizes changes to important parameters, with importance measured using the Fisher Information Matrix. **Online EWC** is a variant that maintains a running estimate of parameter importance, making it more suitable for longer sequences. **MAS** [2] computes importance based on how parameters influence the policy's output, rather than gradients. PackNet [18] incrementally allocates parts of the network to each task through pruning and freezing. Finally, Continual Backpropagation (**CBP**) [9] introduces architectural plasticity by periodically replacing parts of the network to preserve adaptability over many tasks. As the MARL baseline, we opt for IPPO [8]. It is a natural choice as it can be seamlessly integrated with all continual learning methods. It has been shown to outperform other MARL approaches on both SMAC [8] and Overcooked [23], making it a strong candidate for evaluating continual multi-agent learning in a fully observable setting.

### 5.3 Baseline Comparison

Figure 3 compares the performance of several continual learning methods combined with IPPO over a 10-task sequence. None of the methods fully retain prior knowledge, most tasks are completely forgotten once access is lost. Regularization-based approaches like EWC and MAS reduce forgetting to a degree, but their long-term performance gains are limited. PackNet, while somewhat preserving

Figure 4: Comparison of final average performance using MLP vs. CNN encoders for EWC and MAS.

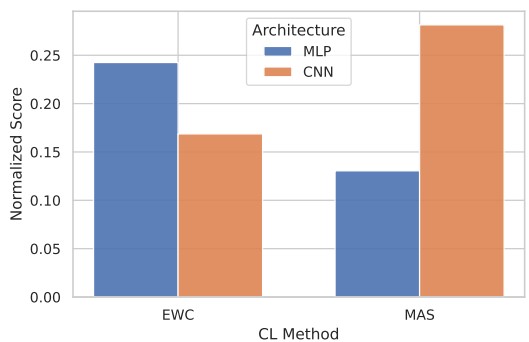

Table 2: Comparison of CMARL performance across CL methods on a 10-task sequence with random order. Results are averaged over 5 seeds. CMARL metrics: $\mathcal{A}$ (avg. performance), $\mathcal{F}$ (forgetting), $\mathcal{P}$ (plasticity).

| Method | $\mathcal{A}\uparrow$ | $\mathcal{F}\downarrow$ | $\mathcal{P}\uparrow$ |
|---|---|---|---|
| FT | 0.122 | 0.682 | 0.794 |
| Online EWC | 0.118 | 0.739 | 0.870 |
| EWC | 0.121 | 0.710 | **0.881** |
| MAS | **0.131** | **0.294** | 0.443 |
| PackNet | 0.040 | 0.318 | 0.312 |
| L2 | 0.064 | 0.695 | 0.735 |

earlier tasks, quickly exhausts its capacity and fails to learn anything. Table 2 reports summary metrics over 5 seeds. MAS achieves the best average performance and lowest forgetting, though at the cost of reduced plasticity. In contrast, methods like FT, EWC and Online EWC display high plasticity but struggle with retention, highlighting the inherent stability–plasticity trade-off in CMARL. These results show the difficulty of maintaining both adaptability and memory in cooperative continual multi-agent environments.

### 5.4 Forgetting

Comparison of CMARL performance across continual learning methods on a 10-task sequence with random order. Results are averaged over 5 seeds. $\mathcal{A}$ measures final average performance, $\mathcal{F}$ captures forgetting, and $\mathcal{P}$ reflects plasticity. MAS achieves the best overall performance and retention, while FT shows high plasticity but suffers from catastrophic forgetting

Figure 5 illustrates the extent of forgetting across tasks for FT, EWC, and MAS. Fine-tune serves as a clear example of catastrophic forgetting. After transitioning to a new task, performance on previous tasks rapidly collapses. In contrast, EWC and MAS manage to retain some knowledge of earlier tasks, particularly the first one, but fail to reach the same training returns on later tasks as FT, demonstrating the trade-off between stability and plasticity.

### 5.5 Encoder Architecture

In our main experiments, we adopt an MLP encoder due to its simplicity and compatibility with low-dimensional inputs. To explore the effect of encoder choice on CMARL, we evaluate EWC and MAS with a CNN-based encoder. Figure 4 shows the impact of architecture on performance. EWC performs slightly better with an MLP encoder, suggesting that its regularization interacts more favorably with simpler representations. In contrast, MAS exhibits a nearly $2\times$, when paired with a CNN encoder, suggesting that its functional sensitivity estimation benefits from spatial structure and richer features.

## 6 Conclusion

We introduced MEAL, the first benchmark for continual multi-agent reinforcement learning. By leveraging JAX for efficient GPU-accelerated training and introducing a diverse set of handcrafted and procedurally generated Overcooked environments, MEAL enables the study of long-horizon continual learning in cooperative settings. Our evaluation of six continual learning methods combined with the IPPO algorithm reveals that existing CL techniques struggle to retain cooperative behaviors while maintaining adaptability to new tasks. Regularization-based methods mitigate forgetting but sacrifice plasticity, while parameter-isolation methods fail to scale with longer task sequences. These findings highlight the need for new approaches that can handle the dual challenges of cooperation and non-stationarity in CMARL. We hope MEAL serves as a foundation for advancing this underexplored but important research direction.

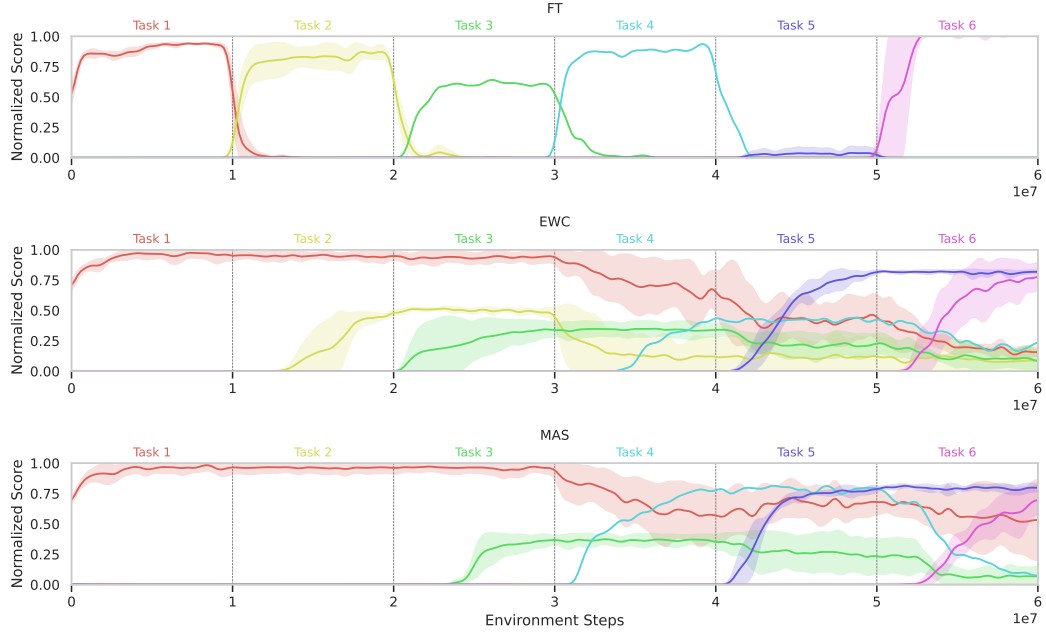

Figure 5: Normalized evaluation score of each task in the 6-task sequence during training.

## 7   Limitations

While MEAL provides a scalable and diverse testbed for continual multi-agent reinforcement learning, several limitations remain. First, MEAL is restricted to fully observable, two-agent environments with discrete action spaces, limiting its applicability to partially observable or competitive multi-agent settings. Second, while layout diversity is high, the domain itself is narrow. Overcooked dynamics do not capture the full complexity of real-world multi-agent interactions involving language, negotiation, or long-horizon planning. Third, our benchmark only evaluates task-incremental learning. Future work could extend MEAL to other continual learning protocols. Finally, we only consider continual learning in settings where the environment layout changes across tasks, but not the partner agent.

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

# A  Environment Layouts

## A.1  Easy Layouts

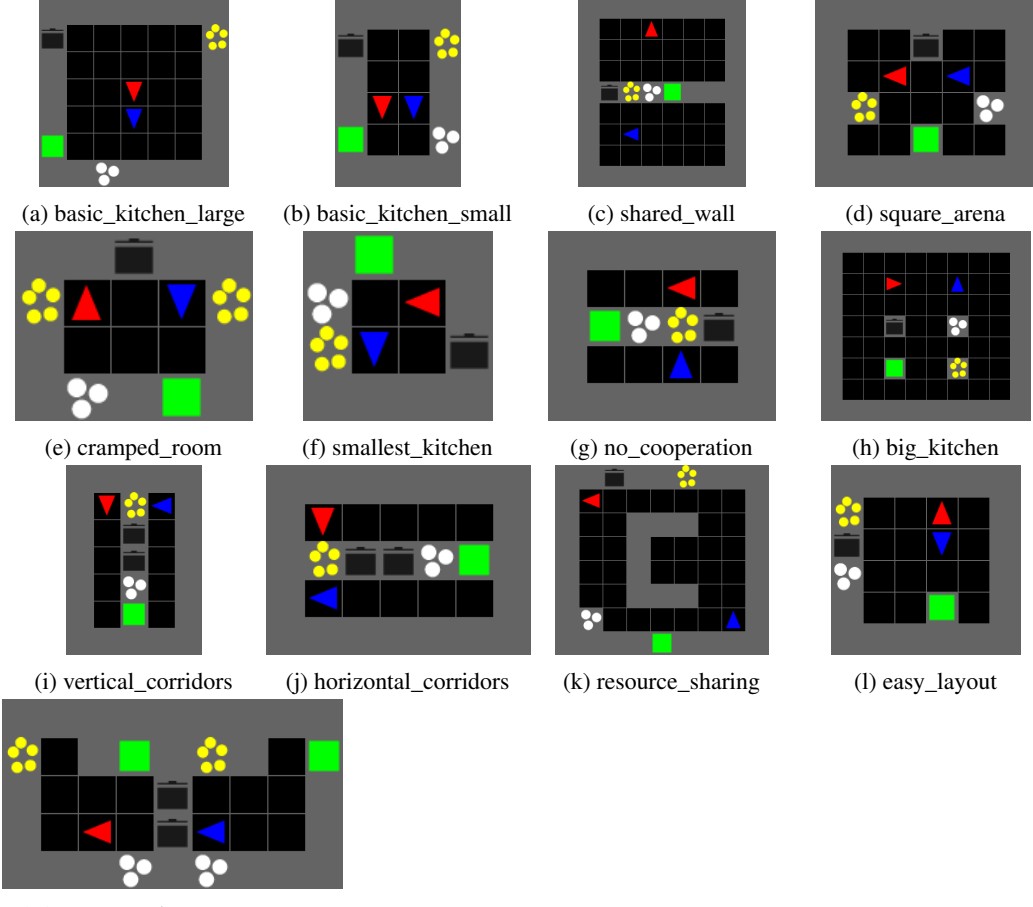

(a) basic_kitchen_large    (b) basic_kitchen_small    (c) shared_wall    (d) square_arena

(e) cramped_room    (f) smallest_kitchen    (g) no_cooperation    (h) big_kitchen

(i) vertical_corridors    (j) horizontal_corridors    (k) resource_sharing    (l) easy_layout

(m) asymm_advantages

Figure 6: Easy MEAL layouts (coordination not required).

## A.2 Medium Layouts

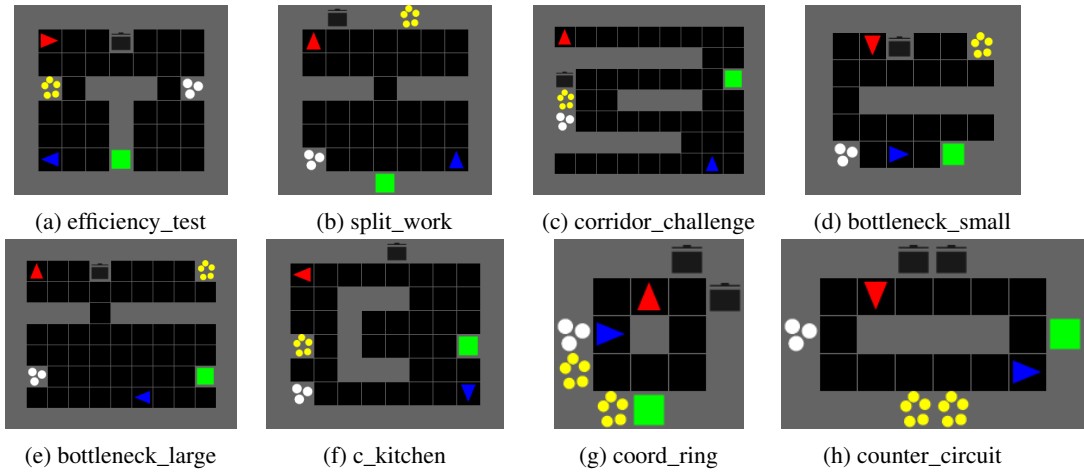

(a) efficiency_test  (b) split_work  (c) corridor_challenge  (d) bottleneck_small

(e) bottleneck_large  (f) c_kitchen  (g) coord_ring  (h) counter_circuit

Figure 7: Medium MEAL layouts (bottlenecks and deadlock risk).

## A.3 Hard Layouts

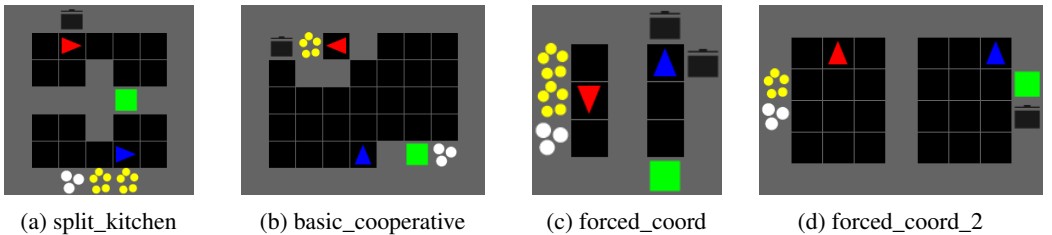

(a) split_kitchen  (b) basic_cooperative  (c) forced_coord  (d) forced_coord_2

Figure 8: Hard MEAL layouts (partitioned regions, specialization needed).

## B Hyperparameters

Table 3: Common hyper-parameters for all MEAL experiments. Values are fixed across methods and experiments unless stated otherwise.

| Parameter | Value |
|---|---|
| *IPPO / optimisation* | |
| Learning rate $\eta$ | $3 \times 10^{-4}$ |
| Anneal LR | No (linear schedule optional) |
| Total env. steps per task $\Delta$ | $10^7$ |
| Num. envs (parallel) | 16 |
| Rollout length T | 128 |
| Update epochs | 8 |
| Minibatches per update | 8 |
| Batch size | $16 \times 128 = 2048$ |
| $\gamma$ | 0.99 |
| GAE $\lambda$ | 0.957 |
| Clipping $\epsilon$ | 0.2 |
| Entropy coef. $\alpha_{\text{ent}}$ | 0.01 |
| Value-loss coef. $\alpha_{\text{vf}}$ | 0.5 |
| Max grad-norm | 0.5 |
| *Continual-learning specifics* | |
| Sequence length $|\mathcal{T}|$ | 10 tasks (random order) |
| CL method coefficients $\lambda$ | $1 \times 10^6$ (EWC) / $1 \times 10^5$ (L2, MAS) |
| EWC mode / decay | Online / 0.9 |
| Importance episodes / steps | 5 / 500 |
| Regularise critic / heads | No / Yes |
| *Misc. settings* | |
| Reward shaping | Yes, anneal to 0 after $2.5 \times 10^6$ steps |
| Evaluation interval | every 100 updates (10 episodes) |
| Seeds | $\{1, 2, 3, 4, 5\}$ |

