# OpenReview forum: "MEAL: A Benchmark for Continual Multi-Agent Reinforcement Learning"
_NeurIPS.cc/2025/Datasets_and_Benchmarks_Track — Submitted to NeurIPS 2025 Datasets and Benchmarks Track_

### Official Review · Reviewer_6HoQ · 2025-06-28

**Rating:** 5
**Confidence:** 5

**Summary:**

This work is the first specially designed benchmark for continual MARL. The author present Multi-Agent Environments for Adaptive Learning (MEAL), a Jax-based Overcooked benchmark with challenging sequential tasks and low computational resource requirement. Through extensive experiments on the proposed benchmark, the authors demonstrate the effectiveness of MEAL and the shortcomings of existing MARL methods.

**Dataset Code Accessibility:**

Yes

**Ethical Considerations:**

No, there are no or only very minor ethics concerns

**Final Justification:**

The authors have addressed my main concerns. I decided to keep my score.

**Limitations Weaknesses:**

1. As presented in the limitation section, the proposed MEAL only include 2 agents, which could reduce the significance of the proposed benchmark in MARL community.

2. Does MEAL support heterogeneous agents? It would be interesting if MEAL support more than 3 heterogeneous agents.

3. In Fig. 3, the curves gradually become stable after environmental changes. However, , it is not clear whether the baselines achieve optimality or fall into sub-optimal regimes, since the maximum return of evaluation is unknown. Thus, it is hard to determine whether the proposed benchmark is learnable and challenging for existing MARL methods.

4. It seems from Fig. 3 that, tested methods show a task-specific performance, *i.e.*, different methods are good at different tasks. Can the author explain the reason behind this results?

**Strengths Contributions:**

The motivation of design a continual MARL benchmark is clear. The significance of the proposed MEAL for MARL community is beyond doubt. The evaluation metrics is reasonable and easy-to-understand.

---

> ### Author Rebuttal · Authors · 2025-07-31
>
> We thank the reviewer for their positive feedback and for recognizing the significance of MEAL as the first dedicated benchmark for CMARL, its clear motivation, and the clarity of our proposed evaluation metrics.
>
> ## Weakness 1: N-Agents
> > *As presented in the limitation section, the proposed MEAL only include 2 agents, which could reduce the significance of the proposed benchmark in MARL community.*
>
> To ensure MEAL remains valuable to the MARL community, we extend the benchmark to support an **N-agent setting**, enabling a systematic study of how increasing the number of cooperating agents impacts CL. In particular, we evaluated EWC combined with PPO in the single-agent case, and IPPO with 2 and 3 agents, on 20-task sequences with shared rewards.
>
> | Agents   | Easy 𝒜↑       | Medium 𝒜↑     | Hard 𝒜↑       | Easy 𝓕↓       | Medium 𝓕↓     | Hard 𝓕↓       |
> |----------|----------------|----------------|----------------|----------------|----------------|----------------|
> | 1 Agent  | 0.62±0.06      | 0.34±0.06      | **0.29±0.23**  | 0.05±0.05      | 0.07±0.04      | 0.16±0.05      |
> | 2 Agents | **0.84±0.03**  | **0.60±0.21**  | 0.18±0.02      | **0.03±0.03**  | **0.06±0.01**  | **0.05±0.06**  |
> | 3 Agents | 0.48±0.36      | 0.28±0.14      | 0.12±0.13      | 0.13±0.08      | 0.13±0.11      | 0.09±0.09      |
>
> We find that a single agent delivers less soup than two because it cannot parallelize tasks: while one agent delivers soup, the other can already load the pot with onions. However, in the hard difficulty, the single-agent PPO outperforms the two-agent IPPO. We observed that, under the continual learning setting, the two-agent setup solved fewer tasks. This appears to be due to the larger grid size in harder layouts: increasing both the observation space and the number of agents makes the learning problem more complex for IPPO, which struggles to develop effective cooperative behaviors in this setting.
>
> Adding a third agent further hurts performance for similar reasons. IPPO trains independent policies while the environment remains a joint MDP, where transitions and rewards depend on the combined actions of all agents. Moving from 2→3 agents expands the joint action space and interaction patterns, amplifies non-stationarity (as two teammates’ policies change simultaneously), and makes credit assignment more difficult (since the reward is shared, IPPO does not know which agent made a good action). Without explicit communication or role allocation, IPPO struggles to learn continually as the team size grows.
>
> We believe extending MEAL to support more agents is a valuable addition, as it increases coordination difficulty, introduces richer role emergence, amplifies interference effects, and adds variability to the task distribution in the continual learning sequences.
>
>
> ## Weakness 2: Heterogeneous Agents
> > *Does MEAL support heterogeneous agents? It would be interesting if MEAL support more than 3 heterogeneous agents.*
>
> We very much appreciate the suggestion. To support a heterogeneous agent setting, we introduce a variant in which agents are assigned predefined roles at the start of each task: **chef** and **server**. The chef is responsible for preparing the soup but cannot pick up plates, while the courier handles dish delivery but cannot pick up onions. This enforces complementary capabilities, meaning neither agent can complete the full recipe alone, and successful play requires coordinated role execution and adaptation.
>
> We compare this variant with the default (homogeneous) setting over a 20-task sequence using EWC+IPPO with shared rewards, measuring average performance (𝒜), forgetting (𝓕), forward transfer (𝓕𝓣), and the area under the training curve (𝓐𝓤𝓒):
>
> | Setting       | 𝒜 ↑         | 𝓕 ↓         | 𝓕𝓣 ↑        | 𝓐𝓤𝓒 ↑       |
> |---------------|-------------|-------------|--------------|-------------|
> | Homogeneous   | 0.90 ± 0.04 | 0.01 ± 0.01 | 0.20 ± 0.08  | 0.50 ± 0.04 |
> | Heterogeneous | 0.68 ± 0.09 | 0.03 ± 0.02 | -0.05 ± 0.09 | 0.38 ± 0.04 |
>
> We observe a clear performance drop in the role-restricted setting, as throughput decreases when agents are limited to certain actions and cannot flexibly switch between tasks. Generalization also degrades because specialization limits the ability to adapt to new role assignments, resulting in weaker forward transfer. In particular, switching agent roles across tasks further reduces transfer, as knowledge learned in one role does not readily apply to a different, complementary role.
>
> ## Weakness 3: Benchmark Challenge
> > *In Fig. 3, the curves gradually become stable after environmental changes. However, , it is not clear whether the baselines achieve optimality or fall into sub-optimal regimes, since the maximum return of evaluation is unknown. Thus, it is hard to determine whether the proposed benchmark is learnable and challenging for existing MARL methods.*
>
> Indeed, the exact optimal performance in MEAL is unknown, as determining the best possible coordination strategy for multiple agents is inherently complex. However, to provide a more meaningful measure of performance, rather than just raw returns, we analytically determine the fastest possible single‑agent cook–deliver cycle for each layout. This yields a normalized score across layouts, where 1.0 corresponds to the single‑agent optimum, and values above 1.0 indicate that multiple agents surpassed solo efficiency through coordination.
>
> From these normalized scores, we compute our core metrics: average performance (𝒜), forgetting (𝓕), and forward transfer (𝓕𝓣). The table below summarizes the results across three difficulty levels on 20-task sequences, averaged over 5 seeds.
>
> | Method     | Easy 𝒜↑ | Medium 𝒜↑ | Hard 𝒜↑ | Easy 𝓕↓ | Medium 𝓕↓ | Hard 𝓕↓ | Easy 𝓕𝓣↑ | Medium 𝓕𝓣↑ | Hard 𝓕𝓣↑ |
> |------------|---------|-----------|---------|----------|-----------|----------|------------|-------------|------------|
> | FT         | 0.05±0.00 | 0.04±0.01 | 0.01±0.02 | 0.90±0.01 | 0.79±0.01 | 0.52±0.08 | 1.79±0.92 | -0.02±1.27 | 7.88±12.92 |
> | EWC        | **0.84±0.03** | **0.60±0.21** | 0.18±0.02 | 0.03±0.03 | **0.06±0.01** | 0.05±0.06 | 0.84±1.90 | -0.84±1.25 | 15.29±32.58 |
> | Online EWC | 0.77±0.09 | 0.59±0.03 | **0.31±0.00** | 0.15±0.09 | 0.21±0.03 | 0.17±0.02 | 1.84±0.82 | 0.21±1.32 | -5.08±9.07 |
> | MAS        | 0.28±0.07 | 0.16±0.09 | 0.03±0.01 | 0.50±0.06 | 0.42±0.07 | 0.25±0.07 | -0.76±0.21 | -0.17±1.93 | **61.41±117.04** |
> | L2         | 0.75±0.02 | 0.50±0.02 | 0.13±0.03 | **0.03±0.00** | 0.11±0.01 | **0.05±0.00** | 0.15±0.62 | -1.83±1.68 | 60.50±116.72 |
> | AGEM       | 0.20±0.05 | 0.12±0.01 | 0.04±0.02 | 0.76±0.07 | 0.63±0.07 | 0.46±0.05 | **2.13±1.73** | **0.34±0.88** | 7.18±12.51 |
>
> At *easy* difficulty, the best methods (EWC and L2) achieve almost zero forgetting, and high average performance, indicating that the setting is indeed learnable. In contrast, under *hard* difficulty, the same methods show much lower performance and higher forgetting, demonstrating that not only do the environments become substantially harder to learn, but that retaining prior knowledge becomes more challenging. These results show that MEAL’s lower‑level tasks are well within reach of current methods, while higher levels expose clear limitations, fulfilling our goal of a benchmark that is both learnable to distinguish existing methods and sufficiently complex to challenge future methods.
>
>
> ## Weakness 4: Task-Specific Performance
> > *It seems from Fig. 3 that, tested methods show a task-specific performance, i.e., different methods are good at different tasks. Can the author explain the reason behind this results?*
>
> We appreciate the reviewer’s observation. Regularization-based methods (e.g., EWC, L2, MAS) have different mechanisms to constrain weight updates of parameters they consider important. This biases learning toward different regions of the parameter space. This bias can align well with future tasks that share representations with past ones but hinder adaptation when tasks differ significantly in layout, coordination demands, or role assignments. Memory-based methods like AGEM can help revisit past tasks but often struggle if replay samples are uninformative for the current task.
>
> Unavoidably, intrinsic RL stochasticity plays a role. In the network plasticity experiment (response to reviewer kbDF), where we repeated training on the same task sequence multiple times, we observed that even the same method can often completely fail to learn a task upon revisiting it, while on another repetition, it performs very well. This variability stems from the stochastic nature of RL. Exploration noise, environment randomness, and high sensitivity to initialization or early trajectory sampling can strongly influence whether the agent discovers a high-reward policy within the given training budget.
>
> We hope this response has addressed the reviewer’s comments and clarified our contributions, and we welcome any further suggestions to improve our work.

---

> > ### Comment · Reviewer_6HoQ · 2025-08-01
> >
> > Thanks for the rebuttal. I think the rebuttal addresses my most concerns. I would keep my score.

---

### Official Review · Reviewer_kbDF · 2025-06-28

**Rating:** 3
**Confidence:** 3

**Summary:**

The author proposed a benchmark named MEAL for continual multi-agent reinforcement learning (CMARL), and used JAX to build the first end-to-end GPU-accelerated task sequences for continual RL. The author implemented six CL methods on MEAL and evaluated them in various environments, revealing the shortcomings of these methods in preserving cooperative behavior and adapting to role changes between tasks.

**Dataset Code Accessibility:**

Yes

**Ethical Considerations:**

No, there are no or only very minor ethics concerns

**Limitations Weaknesses:**

1、In continual multi-agent reinforcement learning (CMARL), there are challenges from both the multi-agent and continual learning perspectives. However, in the evaluation, the authors only focused on the continual learning perspective, while the multi-agent perspective was hardly discussed.
2、The supported CL methods are outdated. The popular methods of recent years have not been mentioned at all.
3、The meanings of different colors, shapes and patterns in Figure should be indicated.
4、 The “CBP” mentioned in line 193 seems not to have been evaluated.

**Strengths Contributions:**

1、It is the first benchmark for continual multi-agent reinforcement learning.
2、This benchmark supports GPU acceleration and has three difficulty levels.
3、The experimental conclusion is somewhat enlightening.

---

> ### Author Rebuttal · Authors · 2025-07-31
>
> We thank the reviewer for their thoughtful feedback and for recognizing the novelty of MEAL as the first CMARL benchmark.
>
> ## Weakness 1: Multi-Agent Perspective
>
> To better analyze the multi-agent dimension, we first extend MEAL to support an N-agent setting, allowing us to systematically study how the number of cooperating agents affects continual learning. We run EWC combined with PPO for the single-agent case, and IPPO for 2 and 3 agents, all on 20-task sequences.
>
> |Agents|Easy 𝒜↑|Medium 𝒜↑|Hard 𝒜↑|Easy 𝓕↓|Medium 𝓕↓|Hard 𝓕↓|
> |------|-------|----------|--------|--------|----------|--------|
> |1 Agent|0.62±0.06|0.34±0.06|**0.29±0.23**|0.05±0.05|0.07±0.04|0.16±0.05|
> |2 Agents|**0.84±0.03**|**0.60±0.21**|0.18±0.02|**0.03±0.03**|**0.06±0.01**|**0.05±0.06**|
> |3 Agents|0.48±0.36|0.28±0.14|0.12±0.13|0.13±0.08|0.13±0.11|0.09±0.09|
>
> A single agent delivers less soup than two because it cannot parallelize tasks: while one agent delivers soup, the other can already load the pot with onions. However, in the hard difficulty, the single-agent PPO outperforms the two-agent IPPO. We observed that, under the CL setting, the two-agent setup solved fewer tasks due to the larger grid size in harder layouts: increasing both the observation space and the number of agents makes the learning problem more complex for IPPO, which struggles to develop effective cooperative behaviors in this setting.
>
> Adding a third agent further hurts performance for similar reasons. IPPO trains independent policies while the environment remains a joint MDP, where transitions and rewards depend on the combined actions of all agents. Moving from 2→3 agents expands the joint action space and interaction patterns, amplifies non-stationarity (as two teammates’ policies change simultaneously), and makes credit assignment more difficult (since the reward is shared, IPPO does not know which agent made a good action). Without explicit communication or role allocation, IPPO struggles to learn continually as the team and layout size grow.
>
> We believe extending MEAL to support more agents is a valuable addition, as it increases coordination difficulty, introduces richer role emergence, amplifies interference effects, and adds variability to the task distribution in the continual learning sequences.
>
>
> Second, following the suggestion of reviewer MF6D, we introduced a **partial observabilty** setting to MEAL, where **MAPPO** [1] is known to outperform IPPO, since its centralized critic can 1) more accurately estimate individual contributions to shared rewards under partial observability, and 2) reduce non-stationarity by conditioning value estimates on the joint actions of all agents, leading to more stable and coordinated policy updates. We investigate this by running a 20-task sequence under partial observability and shared rewards with EWC.
>
> |Algorithm|Easy 𝒜↑|Medium 𝒜↑|Hard 𝒜↑|Easy 𝓕↓|Medium 𝓕↓|Hard 𝓕↓|
> |---------|-------|----------|--------|--------|----------|--------|
> |IPPO|**0.588±0.04**|**0.354±0.07**|**0.186±0.07**|**0.065±0.00**|**0.068±0.01**|**0.040±0.00**|
> |MAPPO|0.299±0.09|0.099±0.01|0.069±0.02|0.126±0.01|0.071±0.01|0.040±0.01|
>
> To our surprise, MAPPO did not outperform IPPO in our experiments. We suspect this may be partly due to suboptimal hyperparameter choices. For IPPO, we adopted parameters from JaxMARL [2], which are well-validated in prior work, while for MAPPO, we did not find established configurations and relied on limited tuning. Additionally, EWC was tuned specifically to work with IPPO and not explicitly optimized for MAPPO. We shall explore this further after this phase of rebuttal.
>
>
> ## Weakness 2: Outdated Baselines
>
> We chose to use established CL baselines rather than SOTA methods. These baselines are widely adopted, better understood, and allow for clearer attribution of continual learning dynamics in CMARL. This mirrors the RL field, where older algorithms such as DQN often remain standard baselines to evaluate certain phenomena despite more recent advances, as they provide a reliable foundation for comparison. Moreover, no explicit CMARL baselines currently exist. Our work is the first to naively combine CL methods with MARL algorithms, laying the groundwork for future, more sophisticated integrations.
>
> Importantly, JAX implementations of most modern CL methods do not exist, and no open-source tools currently support their seamless integration with MARL frameworks. Porting more complex, recent CL methods to JAX and adapting them to multi-agent PPO would require substantial engineering effort beyond the scope of introducing MEAL as a benchmark. Note that even the *outdated* CL methods we include had not been implemented for RL in JAX before.
>
> Regarding this point, we do think that evaluating only regularization-based CL methods would provide a limited perspective. To broaden coverage across different families of CL approaches, we also included AGEM [3], a representative memory-based method, which performed worse than all the regularization-based baselines.
>
> |Method|Easy 𝒜↑|Medium 𝒜↑|Hard 𝒜↑|Easy 𝓕↓|Medium 𝓕↓|Hard 𝓕↓|Easy 𝓕𝓣↑|Medium 𝓕𝓣↑|Hard 𝓕𝓣↑|
> |------|-------|----------|--------|--------|----------|--------|----------|-----------|----------|
> |AGEM|0.20±0.05|0.12±0.01|0.04±0.02|0.76±0.07|0.63±0.07|0.46±0.05|2.13±1.73|0.34±0.88|7.18±12.51|
>
> If there are specific CL methods the reviewer believes should be prioritized for inclusion, we would be happy to consider adding them.
>
> ## Weakness 3: Clarity in Figures
>
> We are not certain which figure(s) the reviewer is referring to, but we assume it is **Figure 5**. In this figure, the color of each line corresponds to the respective task *i*. Each line plots the evaluation performance on task *i* over time.
>
> A general pattern can be observed: the performance on task *i* typically increases when the agent begins training on that task, as it has not been encountered before. If the curve for task *i* remains at the same level after training has moved on to task *i+1*, this indicates that the agent successfully retains the skills learned on task *i*. Conversely, if the curve for task *i* starts to decline after training on task *i* has ended and training has progressed to tasks *i+1, i+2,* and so on, this indicates that the agent is forgetting task *i*, likely due to negative transfer from learning subsequent tasks.
>
> The Fine‑Tuning (**FT**) baseline provides a clear example: the performance for almost every task *i* drops sharply soon after the training on task *i* is finished, demonstrating near‑complete forgetting. We will revise the figure caption to make these interpretations more explicit.
>
>
> ## Weakness 4: CBP
>
> The original motivation for evaluating CBP was to examine **network plasticity**, a well‑known phenomenon in CL, in which neural networks are known to lose their capacity to learn new tasks over time. Our intent was to explore whether such plasticity loss also appears in a **multi‑agent** setting, similar to what has been documented in the single‑agent setting [4, 5, 6].
>
> However, solely comparing baselines (e.g., FT vs. CBP) in isolation would offer limited insight. We therefore adopted a more informative approach: running IPPO continually over repeated task sequences without resetting the network weights, to directly quantify plasticity loss over time.
>
> Specifically, we train on a 10‑task sequence and repeat it for 3 iterations and 10 iterations. For each task and each repetition, we track the following commonly used metrics, then average them across all tasks in the sequence:
> - AUC‑loss (↓) – Normalized drop in area under the learning curve compared to the first pass, capturing how much learning capacity is lost on revisited tasks.
> - FPR (↑) – Ratio of final performance in later repetitions to the first, indicating how well peak performance is retained.
> - RAUC (↑) – Ratio of overall area under the curve in later repetitions to the first, reflecting how much of the original learning ability is preserved throughout training.
>
> This setup isolates plasticity loss, the gradual inability to fit new data under continual training.
>
> |Repeats|AUC‑loss↓|FPR↑|RAUC↑|
> |-------|---------|----|-----|
> |1|0.000±0.000|1.000±0.000|1.000±0.000|
> |3|0.154±0.135|0.906±0.188|0.905±0.164|
> |10|0.215±0.100|0.866±0.116|0.846±0.120|
>
> We observe that all metrics deteriorate with more repetitions, indicating that plasticity loss is indeed present in the multi-agent setting. AUC‑loss increases by roughly 40% when going from 3 to 10 repetitions. FPR is already below 1 even after 3 repetitions, showing some loss of final performance early on. Interestingly, even though our setting spans over 1B environment steps, well beyond the scale of prior studies, those works report a much stronger loss of plasticity than observed in MEAL. We hypothesize that this difference stems from our experiments using multiple output heads, which isolate task-specific outputs, reduce gradient interference, and preserve prior policies while allowing the backbone to learn transferable features.
>
> [1] Yu, Chao, et al. "The surprising effectiveness of ppo in cooperative multi-agent games." Advances in neural information processing systems 35 (2022): 24611-24624.
>
> [2] Rutherford, Alexander, et al. "Jaxmarl: Multi-agent rl environments and algorithms in jax." Advances in Neural Information Processing Systems 37 (2024): 50925-50951.
>
> [3] Chaudhry, Arslan, et al. "Efficient lifelong learning with a-gem." arXiv preprint arXiv:1812.00420 (2018).
>
> [4] Kirkpatrick, James, et al. "Overcoming catastrophic forgetting in neural networks." Proceedings of the national academy of sciences 114.13 (2017): 3521-3526.
>
> [5] Zenke, Friedemann, Ben Poole, and Surya Ganguli. "Continual learning through synaptic intelligence." International conference on machine learning. PMLR, 2017.
>
> [6] Aljundi, Rahaf, et al. "Online continual learning with maximal interfered retrieval." Advances in neural information processing systems 32 (2019).

---

> > ### Author Response · Authors · 2025-08-06
> >
> > Dear Reviewer kbDF,
> >
> > As the discussion phase is approaching its end, we wanted to follow up and see if you have had a chance to review our rebuttal. We have done our best to address all the concerns you have raised, and we would greatly appreciate any further feedback you might have, or let us know if there is anything else we should clarify or add before the deadline.
> >
> > Thanks again for your time and for reviewing our work!
> >
> > Best regards,
> > The authors

---

### Official Review · Reviewer_MF6D · 2025-07-03

**Rating:** 4
**Confidence:** 3

**Summary:**

This paper introduces MEAL, a benchmark designed for studying Continual Multi-Agent Reinforcement Learning (CMARL). MEAL is built on the JaxMARL version of the Overcooked environment, a cooperative multi-agent task with complex credit assignment challenges. The benchmark focuses on task-incremental continual learning, offering multiple layout variations to simulate environment shifts. Leveraging JAX and GPU acceleration, MEAL provides efficient simulation.

**Dataset Code Accessibility:**

Yes

**Dataset Code Comments:**

The code is available at the link provided by the author.

**Ethical Considerations:**

No, there are no or only very minor ethics concerns

**Final Justification:**

The authors have addressed some of my concerns in the revision, particularly by incorporating Partial Observability and adding additional experiments. These improvements have strengthened the paper, and as a result, I am raising my score to a borderline accept.

**Limitations Weaknesses:**

1. The task diversity in MEAL is currently limited—primarily varying layout configurations in Overcooked. This may not fully test continual learning capabilities. Adding variation in reward structures, agent types, or task goals would strengthen the benchmark.
2. The environments appear to be fully observable, which reduces the realism of multi-agent coordination challenges. Many real-world scenarios involve partial observability, and extending MEAL to include partial views (e.g., occlusion, field-of-view limits) would improve its relevance.
3. It is unclear whether the benchmark samples layouts across all three difficulty levels or within a fixed level. Explicitly supporting curriculum learning (e.g., easy to hard) would be a valuable extension.
4. MEAL’s core environments are quite similar to standard Overcooked settings. The paper should more clearly highlight what is new.

**Strengths Contributions:**

1. The paper studies an important and underexplored problem at the intersection of continual learning and multi-agent RL, with relevance to lifelong learning in collaborative AI systems.
2. The benchmark is based on the Overcooked task/environment, which provides a challenging and strong testbed to study CMARL.
3. MEAL leverages JAX for GPU acceleration, enabling fast environment rollouts and improved scalability compared to similar frameworks.
4. Overall, the paper is well-structured, easy to understand.

---

> ### Author Rebuttal · Authors · 2025-07-31
>
> We thank the reviewer for their constructive feedback and for recognizing the strengths of our work.
>
> ## Weakness 1: Task Diversity
> > *The task diversity in MEAL is currently limited—primarily varying layout configurations in Overcooked. This may not fully test continual learning capabilities. Adding variation in reward structures, agent types, or task goals would strengthen the benchmark.*
>
> We appreciate the reviewer’s suggestions. We agree that task diversity is limited. As a first step, we have switched to using procedurally generated environments rather than a fixed set of hand‑designed layouts, to achieve better layout diversity. The difficulty level now directly dictates the kitchen width, height, and wall density (for example, Level 2 uses widths and heights sampled from 8–9 tiles with a fixed wall density of 25%). The generator first draws an empty grid, randomly places interactive stations, adds walls according to the specified density, prunes unreachable tiles, and finally validates that the resulting map is solvable. This process yields an effectively unbounded stream of possible layouts.
>
> That said, increased layout variation alone does not fully address the concern. To introduce further diversity, we also include alternative reward settings: a sparse‑reward mode, where agents only receive a shared reward upon successful delivery, and an individual‑reward mode, which emphasizes the sequential social‑dilemma aspect by assigning each agent its own reward signal. We evaluate this setting on IPPO+EWC on the 10-task sequence of *easy* difficulty over 5 seeds, measuring Average Performance (𝒜), Forgetting (𝓕), and Forward Transfer (𝓕𝓣).
>
> | Reward Setting | 𝒜 ↑ | 𝓕 ↓ | 𝓕𝓣 ↑ |
> |----------------|------|------|-------|
> | Shared (Default)| 0.90 ± 0.04 | 0.01 ± 0.01 | 0.20 ± 0.08 |
> | Individual | 0.84 ± 0.08 | 0.08 ± 0.07 | 0.12 ± 0.06 |
> | Sparse     | 0.19 ± 0.04 | 0.02 ± 0.01 | -0.79 ± 0.10 |
>
> The original shared reward setting ranks best in all metrics. The individual reward setting occasionally allows the agents to find a better global solution due to the inherent competitiveness. Agents are motivated to use different onion piles and pots to maximize their own rewards, which often leads to a more efficient solution. However, this competitive drive can also prevent them from converging on a stable solution. The sparse reward setting only grants rewards for successful deliveries, so without a targeted exploration mechanism, it is unlikely even on Level 1 layouts for agents to discover a full delivery sequence through random actions, leading to notably worse performance.
>
> To further diversify tasks by agent types, we introduce a variant where the two agents have complementary restrictions: one cannot pick up onions, while the other cannot pick up plates. This effectively assigns them distinct roles, randomly chosen at the start of each task. Successful play now requires coordinated role execution and adaptation, since neither agent can complete the full recipe alone: one must make the soup while the other has to deliver it.
>
> | Setting    | 𝒜 ↑         | 𝓕 ↓         | 𝓕𝓣 ↑        | 𝓐𝓤𝓒 ↑       |
> |------------|-------------|-------------|--------------|-------------|
> | Default    | 0.90 ± 0.04 | 0.01 ± 0.01 | 0.20 ± 0.08  | 0.50 ± 0.04 |
> | Restricted | 0.68 ± 0.09 | 0.03 ± 0.02 | -0.05 ± 0.09 | 0.38 ± 0.04 |
>
> We observe a performance drop in this setting, as the agents cannot reach the same level of recipe throughput. Moreover, generalization suffers because each agent becomes more specialized, making it harder to adapt to the other role.
>
> ## Weakness 2: Partial Observability
> > *The environments appear to be fully observable, which reduces the realism of multi-agent coordination challenges. Many real-world scenarios involve partial observability, and extending MEAL to include partial views (e.g., occlusion, field-of-view limits) would improve its relevance.*
>
> Thank you for the helpful suggestion. We agree that partial observability adds realism and complexity to multi-agent coordination, and we have now implemented a partially observable version of Overcooked in MEAL. To simulate limited perception similar to real-world agents, we restrict each agent's field of view, defining a rectangular local observation grid centered around the agent, occluding tiles outside the view, and enforcing a more realistic egocentric perspective and necessitating exploration and memory. The size of the observation window scales with difficulty (see table below).
>
> | Difficulty | Grid Size         | Forward View | Side View  | Rear View   | Obs Window (H×W) |
> |------------|-------------------|--------------|------------|-------------|------------------|
> | Easy       | 6–7               | 1            | 1          | 0           | 2×3              |
> | Medium     | 8–9               | 2            | 1          | 0           | 3×3              |
> | Hard       | 10–11             | 3            | 2          | 1           | 5×5              |
>
> We run IPPO+EWC on a 20-task sequence in both the fully observable (FO) and partially observable (PO) settings. All other settings remain identical.
>
> | Observability | Easy 𝒜↑      | Medium 𝒜↑    | Hard 𝒜↑      | Easy 𝓕↓      | Medium 𝓕↓    | Hard 𝓕↓      |
> |---------|---------------|---------------|---------------|---------------|---------------|---------------|
> | Full (FO)      | **0.839±0.03**| **0.604±0.21**| 0.178±0.02    | **0.031±0.03**| **0.061±0.01**| 0.053±0.06|
> | Partial (PO)      | 0.588±0.04    | 0.354±0.07    | **0.186±0.07**| 0.065±0.00    | 0.068±0.01    | **0.040±0.00**    |
>
>
> Partial observability notably increases task difficulty, reducing average performance (𝒜) and slightly increasing forgetting. This confirms the added challenge of coordinating with incomplete information, while still allowing some generalization. Interestingly, the PO setting leads to slightly lower forgetting (𝓕) at higher difficulty, potentially due to increased stochasticity acting as a form of implicit regularization. These results support the inclusion of PO in MEAL for more realistic continual MARL benchmarking.
>
>
> ## Weakness 3: Curriculum Learning
> > *It is unclear whether the benchmark samples layouts across all three difficulty levels or within a fixed level. Explicitly supporting curriculum learning (e.g., easy to hard) would be a valuable extension.*
>
> In the current version, the difficulty level is specified at the start of an experiment, and the entire task sequence is composed of procedurally generated layouts that match that chosen level. Layouts are not mixed across levels within a single run.
>
> We agree that a training curriculum would be valuable for MEAL. We therefore design a setting where each difficulty level contributes an equal number of tasks to the sequence. In our experiments, we instantiate this by using five layouts per level, progressing from easy (1-5) to medium (6-10) to hard (11-15). We measure the average performance (𝒜) metric on tasks in the later stages of training to see if curriculum learning improves retention and adaptation.
>
> | Strategy   | Medium (6–10) 𝒜 ↑  | Hard (11–15) 𝒜 ↑   |
> |------------|--------------------|---------------------|
> | Default    | 0.693 ± 0.147       | 0.328 ± 0.238        |
> | Curriculum | 0.668 ± 0.152       | **0.653 ± 0.181**    |
>
> The results show no statistically significant difference between the default and curriculum strategies on Level 2, given the high variance. However, on Level 3, the curriculum strategy achieves substantially higher performance. A possible explanation is that, under curriculum training, the agent first experiences 5 easy and 5 medium environments, where it receives denser reward signals and more frequent successes. This exposure likely builds useful priors and stabilizes learning, improving adaptation to harder tasks later. In contrast, the default strategy trains only on hard tasks throughout, where learning is more challenging and rewards are much sparser, leading to weaker performance overall.
>
> ## Weakness 4: Novelty
> > *MEAL’s core environments are quite similar to standard Overcooked settings. The paper should more clearly highlight what is new.*
>
> We agree that relying on a limited set of fixed Overcooked layouts offers limited diversity and novelty. To address this, we have switched to using procedurally generated environments instead of hand‑crafted maps.
>
> The generator samples a layout width and height from a range, dictated by the difficulty level, builds an empty grid with outer walls, randomly places interactive stations (onion piles, pots, plates, delivery counters), adds additional walls to reach the target obstacle density (dictated by the difficult level), samples agent starting positions, and finally replaces unreachable floor tiles with walls.
>
> Then, a validator ensures that the generated layout is solvable. It runs a DFS from each agent’s start position to verify that all key object families (onions, pots, plates, delivery counters) are reachable by at least one agent, and that the other agent can meaningfully contribute (for example via hand‑offs across connected regions). If a full cook‑deliver path does not exist, the process restarts until producing a valid layout.
>
> This enables a virtually unlimited supply of diverse layouts, which is essential for long continual learning task sequences, one of the core motivations of our work. In response to reviewer kbDF, we evaluate loss of plasticity across 100 tasks, demonstrating the utility of this feature. We retain the original hand‑crafted layouts as optional evaluation environments, as these fixed maps can be valuable for demonstrating specific skills that agents have learned.
>
> We hope this response sufficiently addresses the reviewer's concerns and strengthens our contributions. Please let us know if any remaining points require further attention.

---

> > ### Author Response · Authors · 2025-08-06
> >
> > Dear Reviewer MF6D,
> >
> > As the discussion phase is approaching its end, we wanted to follow up and see if you have had a chance to review our rebuttal. We have done our best to address all the concerns you have raised, and we would greatly appreciate any further feedback you might have, or let us know if there is anything else we should clarify or add before the deadline.
> >
> > Thanks again for your time and for reviewing our work!
> >
> > Best regards,
> > The authors

---

> > ### Comment · Reviewer_MF6D · 2025-08-07
> >
> > Thank you to the authors for their rebuttal. The authors have addressed some of my concerns in the revision, particularly by incorporating Partial Observability and adding additional experiments. These improvements have strengthened the paper, and as a result, I am raising my score to a borderline accept.

---

> > > ### Author Response · Authors · 2025-08-07
> > >
> > > Dear Reviewer MF6D,
> > >
> > > Thank you for revisiting our paper and for raising your score. We’re glad the revisions helped. If there’s anything else we could improve or clarify before the deadline, we’re happy to do so.
> > >
> > > Best regards,
> > > The authors

---

### Comment · Area_Chair_JfxM · 2025-08-03

Dear reviewers,

Thank you for agreeing to review at NeurIPS 2025. This is a friendly reminder that the authors have submitted their rebuttal and we are now in author-discussion phase until **August 6th AoE**.

Please **acknowledge reading the rebuttal** if you have not done it yet and update your score accordingly. Keep the deadline in mind when asking the authors for further clarification so they have **enough time to reply**.

Thank you,

AC

---

### Decision · Program_Chairs · 2025-09-18

**Decision:**

Reject

**Comment:**

Overall
=====
This work fills an existing gap in benchmarks for Continual Multi-Agent Reinforcement Learning (CMARL). Initial reviews were 3, 3, 5, however, the authors did a good job with the rebuttal, increasing to 4, 3, 5. While the two reviewers that engaged are inclined towards acceptance, kbDF (3) did not engage in the reviewing process and I believe the authors did resolve the reviewer's concerns. Thus I recommend this work to be accepted with a score of 4, 4, 5.

Summary
=======
This submission introduces MEAL (Multi-Agent Environments for Adaptive Learning), the first benchmark specifically designed for Continual Multi-Agent Reinforcement Learning (CMARL). Built on JAX and the Overcooked environment, MEAL provides GPU-accelerated simulation of cooperative multi-agent tasks with varying difficulty levels and layout configurations. The authors evaluate six continual learning methods combined with multi-agent RL algorithms, revealing significant challenges in preserving cooperative behaviors and adapting to environmental changes across sequential tasks.

Strengths
=======
* Novel contribution: First dedicated benchmark for continual multi-agent RL, addressing an important and underexplored intersection of two active research areas
* Technical implementation: JAX-based GPU acceleration enables efficient large-scale experiments with procedurally generated environments across multiple difficulty levels
* Extensive evaluation: Systematic analysis of continual learning methods in multi-agent settings, revealing unique challenges like coordination degradation and role adaptation difficulties

Weaknesses
=========
Beyond requesting clarifications the main weaknesses as reported by reviewers were:
* MF6D: limited task diversity, assuming full observability
* kbDF: too much focus on CL and lack of focus on multi-agent dimension. Outdated CL citations.
* 6HoQ: was concerned about the limited number of agents

The authors did a honest effort to address these concerns adding partial observability, different reward modes, and increasing the number of agents. As a result MF6D increased their score to 4 and 6HoQ kept their score (5). kbDF was flagged for insufficient review and failed to engage during the rest of the process. Since I consider the rebuttal did solve kbDF's concerns, I am assuming a score of 4 instead of 3 for this reviewer.

===== FINAL UPDATE FROM DB Track PCs ====

The final decision for this paper has been taken by the program chairs after consultation with the SACs. All Senior Area Chairs have ranked papers according to the feedback from the AC during the review process. We decided to leave the original meta-review to reflect the opinion of the AC in light of the initial discussions with reviewers and SAC.